# ACCELERATED ONLINE REINFORCEMENT LEARNING USING AUXILIARY START STATE DISTRIBUTIONS

## ABSTRACT

Learning a robust policy that is performant across the state space, in a sample efficient manner, is a long-standing problem in online reinforcement learning (RL). This challenge arises from the inability of algorithms to explore the environment efficiently. Most attempts at efficient exploration tackle this problem in a setting where learning begins from scratch, without prior information available to bootstrap learning. However, such approaches often fail to fully leverage expert demonstrations and simulators that can reset to arbitrary states. These affordances are valuable resources that offer enormous potential to guide exploration and speed up learning. In this paper, we explore how a small number of expert demonstrations and a simulator allowing arbitrary resets can accelerate learning during online RL. We show that by leveraging expert state information to form an auxiliary start state distribution, we significantly improve sample efficiency. Specifically, we show that using a notion of safety to inform the choice of auxiliary distribution significantly accelerates learning. We highlight the effectiveness of our approach by matching or exceeding state-of-the-art performance in sparse reward and dense reward setups, including with images states spaces, even when competing with algorithms with access to expert actions and rewards. Moreover, we find that the improved exploration ability facilitates learning more robust policies in sparse reward, hard exploration environments.

## 1 INTRODUCTION

Online reinforcement learning algorithms learn general behaviors without inductive biases and domain expertise through trial and error. By learning from environmental interaction, such methods can potentially exceed the performance of supervised learning alternatives, reaching superhuman performance levels on tasks such as Atari (Mnih et al., 2015) and Go (Silver et al., 2016). Despite such successes, exploring environments efficiently remains challenging, resulting in long training times (Pathak et al., 2017; Ecoffet et al., 2021; Song et al., 2023).

There has been a considerable amount of work on making online RL more efficient by promoting exploratory behaviors that are novelty-seeking (Pathak et al., 2017) and state space-covering (Haarnoja et al., 2018; Jain et al., 2023a; Seo et al., 2021). Although such approaches have the potential to learn robust policies, the lack of task-directed exploratory cues (Mehta et al., 2022) and a tendency to forget how to revisit promising exploration frontiers (Ecoffet et al., 2021) make them inefficient at learning to solve hard-exploration tasks. Moreover, these methods have been designed to improve exploration efficiency without prior information. Consequently, when expert data or a simulator with arbitrary reset conditions are available, these approaches fail to adequately leverage these additional resources to accelerate exploration.

In contrast to conventional RL, imitation learning (Ho & Ermon, 2016) and offline RL (Kostrikov et al., 2022; Kumar et al., 2020) can learn task-specific behavior purely from offline data. These methods perform well within the distribution of the training data but fail to be robust in an out-of-distribution (OOD) setting, making them unsuitable for real-world applications where a sim2real gap is present. Moreover, they require access to expert actions and rewards, which is not always available in practice.

Hybrid RL approaches mix offline data with online interactions to bridge this gap and learn robust policies efficiently. Bootstrapping online training with offline data is not straightforward, and

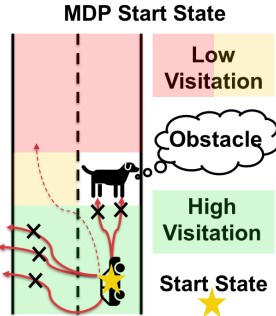 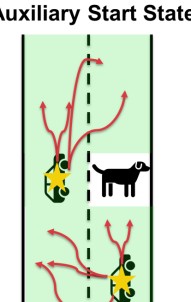

Figure 1: This illustrates a simple driving scenario where a vehicle must learn to go around an object blocking its path. On the left, training begins from the MDP start state. Episode terminations due to collisions and offroad infractions can make visiting states besides and beyond the object difficult. The auxiliary start state distribution on the right alleviates this issue by enabling a more uniform visitation of states during training allowing the policy to simultaneously learn to act before, besides and beyond the object.

naively fine-tuning a policy learned offline often leads to sub-optimal performance (Uchendu et al., 2023). In particular, offline experience can be quickly forgotten during online training if not handled appropriately (Uchendu et al., 2023; Song et al., 2023). Successful hybrid methods ensure the persistence of offline data during online training by freezing a part of the replay buffer or by learning fixed reference policies using the offline data. However, like imitation learning and offline RL, these methods rely on access to expert action and reward information.

In this paper, we revisit the hybrid RL setup and investigate how limited quantities of expert offline data can be used to bootstrap online RL effectively. More specifically, by using expert offline data to construct auxiliary start state distributions, we accelerate online learning considerably, provided the environment can be reset to arbitrary states.

To summarize the main contributions of this work -

- We show that when an arbitrarily resetable simulator is available we can use a small amount of state information collected from an offline expert to create an auxiliary start state distribution that significantly improves the sample-efficiency of online RL, particularly in sparse reward, hard exploration problems including those with image state spaces.

- We find that using a notion of safety, approximated via episode length information, is crucial for forming auxiliary start state distributions that accelerate training. Moreover, we show that this yields policies more robust to shifts in the start state distribution.

- We empirical highlight our findings through performance matching or exceeding competing methods on dense and sparse reward continuous tasks without requiring access to costly affordances such as actions and rewards.

## 2 RELATED WORK

We explore related literature in this space through three broad category of methods: i) purely online RL, ii) purely offline learning and iii) hybrid RL methods.

**Exploration in purely online RL:** Exploration is an age-old problem in reinforcement learning that has received significant attention in the online RL context. Several methods inject additive noise to the actions (Schulman et al., 2017) or network parameters (Burda et al., 2019) to perform exploration. Such exploration is incidental to the primary objective of reward maximization and not very efficient at exploring the state space (Song et al., 2023). Many approaches incentivize exploratory behaviour through exploration bonuses such as surprise-maximizing intrinsic motivation (Pathak et al., 2017), surprise-minimizing intrinsic motivation (Berseth et al., 2021), and action (Haarnoja et al., 2018), state (Seo et al., 2021) and trajectory (Jain et al., 2023b) entropy maximizing rewards. Entropy maximization approaches fail to distinguish exploration in unseen regions from

exploration in regions of the state space where the policy is already proficient. This makes them inefficient. While intrinsic motivation based methods are guided by a notion of surprise, they too struggle in hard-exploration sparse-reward environments (Ecoffet et al., 2021). Moreover, these methods are unable to leverage affordances like offline data and resetable simulators when available.

Go-explore (Ecoffet et al., 2021) is a conceptual framework that disentangles the question of where to explore from how to get there. It is reminiscent of classical planners that first choose an exploration frontier, navigate to it quickly without exploring (or by resetting the simulator to that frontier state) and then initiate exploration after arriving at the frontier. Go-explore maintains an archive of visited states and chooses an exploration frontier from this archive either uniformly at random or using a domain specific heuristic. Our work follows a similar theme but extends this framework by investigating what is a good way of picking an exploration frontier. We present generic properties that are desirable to have in this selection procedure and present a mechanism to select exploration frontiers that will be broadly applicable across a range of tasks.

BARL (Mehta et al., 2022) is an information theoretic exploration method that uses a classical planner and a learnt posterior model to sample transitions that are maximally informative for the policy to learn a given task. This enables it to solve tasks very efficiently. The setting used by its authors bares close resembles to ours since they assume access to a simulator that supports arbitrary resets. Moreover, their use of a Gaussian process (GP) to model the posterior is amenable to utilizing expert demonstrations during training. While very effective on small scale problems with dense reward functions, BARL unfortunately does not scale to higher dimensions and sparse-reward settings. This is confirmed by us in our experiments.

**Learning policies efficiently offline:** Another way to efficiently learn policies is by training on a purely offline dataset of experiences. This sidesteps the issue of online exploration and efficiently recovers a policy based on the offline dataset. Methods such as behaviour cloning and GAIL (Ho & Ermon, 2016) fall under the broad class of imitation learning algorithms that model policy learning as a supervised learning problem and learn a mapping from states to actions. A key issue with imitation learning methods is that they are highly brittle and require access to large amounts of high quality expert data to succeed (Rashidinejad et al., 2021).

Offline reinforcement learning (Levine et al., 2020) is another offline learning paradigm capable of efficiently learning policies from demonstration data of mixed quality while requiring good state coverage in the offline dataset (Kumar et al., 2022; Rashidinejad et al., 2021). Consequently, a key challenge with this approach is that the lack of online interactions leaves offline RL susceptible to distribution shift. Wrongly estimating values for actions beyond the support of the dataset can hamper training (Levine et al., 2020). Conservative Q-learning (CQL) (Kumar et al., 2020) is a recent offline RL method that attempts to tackle this problem by maintaining pessimism within the Q-value function towards actions that are absent from the offline dataset. Implicit Q-learning (IQL) (Kostrikov et al., 2022) completely avoids predicting value estimates for unseen actions by learning a distributional state-value function and computing an upper expectile over it to obtain the value estimate of the best action in that state. These algorithms invariably encounter OOD states and actions in-the-wild, quantities that such algorithm are not robust to by design, making online finetuning a necessity for their real world deployment.

**Hybrid Reinforcement Learning:** Hybrid reinforcement learning leverages a combination of offline data with online interaction to learn policies. The main challenge in hybrid reinforcement learning is to devise methods that effectively bootstrap online learning from offline data. Several approaches (Rajeswaran et al., 2018; Hester et al., 2018) do this by using imitation to learn a policy from offline demonstrations before finetuning it with RL. However, most modern state-of-the-art online RL algorithms are value based (Haarnoja et al., 2018; Schulman et al., 2017). Naively finetuning an offline acquired policy with value based RL can cause significant performance degradation as a value function of similar quality to the pretrained policy is not available at the start of online finetuning (Uchendu et al., 2023). Though Monte Carlo return estimate based algorithms exist, their online finetuning is known to be less efficient (Nair et al., 2020). Offline RL presents a transferable paradigm to train a policy and value function with identical objectives in both offline and online setups. However, not all offline RL methods are well suited for online finetuning due to their inherent pessimism towards distribution shift (Nair et al., 2020). Even better suited offline RL methods like IQL (Kostrikov et al., 2022) result in weaker policies after finetuning especially when limited offline data is available (Uchendu et al., 2023). An alternative line of work (Song et al., 2023; Nair

et al., 2018; Vecerik et al., 2017) avoids finetuning a pretrained policy all together by pre-filling replay buffers at the start of training with transitions from the offline dataset. These transitions persist through training and policy learning happens from scratch. JSRL (Uchendu et al., 2023) presents an alternative approach to the finetuning-free idea of hybrid RL. It learns a guide policy from offline data and uses it to roll out a part of the online episode before handing over control to a freshly initialized policy for completing the roll-out. The handover point is altered over the course of training and all the captured experience is used to train the freshly initialized policy.

The proposed work also lies in this finetuning-free hybrid setting and shares similarities with JSRL. Both the proposed work and JSRL conceptually belong to the Go-explore (Ecoffet et al., 2021) family of algorithms. The two key differences between JSRL and the proposed work are: i) both works encapsulate the idea of a reset distribution or equivalently a frontier state to explore from. While ours is reached through environmental resets, JSRL uses a guide policy to reach it. For a fair comparison, we ignore the time it takes JSRL's guide policy to reach the handover point and perform comparisons for the rollout beyond the handover point. ii) More importantly, JSRL induces a specific kind of reset distribution through its variation of the handover point over the course of training. This differs from our reset distribution and we will show through our experiments that this is an important choice that influences the sample-efficiency and performance of the learnt policy.

## 3 PRELIMINARIES

We define a finite-horizon discrete-time MDP $\mathcal{M}$ to be a $(\mathcal{S}, \mathcal{A}, r, p_0, H, \mathcal{T}, \gamma)$ tuple where $\mathcal{S}$ is the state space, $\mathcal{A}$ is the action space, $r : \mathcal{S} \times \mathcal{A} \times \mathcal{S} \to \mathbb{R}$ is the reward function, $p_0$ is a probability distribution defined over $\mathcal{S}$ corresponding to the start state distribution of $\mathcal{M}$, $H \in \mathbb{N}$ is the finite time horizon, $\gamma$ is the discount factor and $\mathcal{T} : \mathcal{S} \times \mathcal{A} \to P(\mathcal{S})$ describes a transition function capturing the distribution of next states when an action $a \in \mathcal{A}$ is taken at a state $s \in \mathcal{S}$.

In RL, the goal is to obtain a policy $\pi(a|s)$ that maximizes the expected sum of future discounted rewards from $p_0$. Concretely, the objective is to maximize

$$\mathcal{J}_{p_0}(\pi) = \mathbb{E}_{s_0 \sim p_0, s_{t+1} \sim \mathcal{T}(s_t, a_t), a_t \sim \pi(s_t)}[\Sigma_{t=0}^{H} \gamma^t r(s_t, a_t, s_{t+1})] \tag{1}$$

Kakade & Langford (2002) have shown that training on $p_0$ can down-weight the influence of unlikely but important states during policy improvement by visiting them infrequently. Informally, these states, which we refer to as *task-critical states* ($\mathcal{C}$), are a set of states that would be a part of each trajectory from a start state distribution $p_0$ under an optimal policy $\pi^*$ but have a low likelihood under the state visitation distribution of an arbitrary policy $\pi$ that is active during training (yellow region in Figure 1). Thus, to ensure steady learning progress it is important to improve the visitation of $\mathcal{C}$ during training. As a result, the policy can quickly learn good actions for these states and utilize its training budget in propagating policy improvements to other more easily explorable parts of the state space. For example in Figure 1, this could facilitate learning to go around the obstacle from a variety of different starting orientations and velocities, beyond what is captured by $p_0$ (depicted by the star in Figure 1). The objective in Equation 3 does not inherently capture this robustness enhancement. Following the findings of Rajeswaran et al. (2017) where they highlight that training from a diverse set of starts helps learn more robust policies, we define $\mathcal{J}_{\mu_{OOD}}$ to be the expected reward from a different start state distribution $\mu_{OOD}$ which comprises states that would be out-of-distribution (OOD) with respect to $p_0$ (full equation in Appendix 7.2). A robust sample-efficient policy would not only quickly learn to maximize $\mathcal{J}_{p_0}$ but would also improve robust performance as measured by $\mathcal{J}_{\mu_{OOD}}$. For policies designed to operate in the real-world, it is important for them to generalize beyond the training distribution, making evaluation from $\mu_{OOD}$ a useful benchmark to consider while training Rajeswaran et al. (2017).

## 4 AUXILIARY START STATES FOR ACCELERATED LEARNING

Directly computing the visitation distribution over *task-critical states* ($\mathcal{C}$) is computationally infeasible and challenging to approximate. Moreover, as this is a policy-dependant quantity, it needs to be continuously recomputed over the course of training. As a result, a suitable $\mu$ should be a dynamic distribution that is easy to compute and accounts for the policy-induced changes in the visitation distribution.

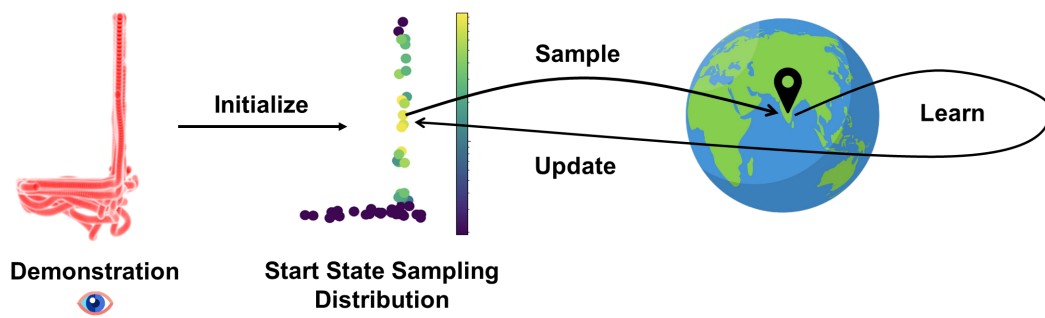

Figure 2: This figure provides an overview of the proposed approach. State information from the offline expert demonstrations is used to initialize a start state sampling distribution. During training start states are drawn from this distribution, used to generate rollouts, train on them and finally update the sampling probabilities (depicted by the color spectrum in the middle) based on the length of the rollout.

We observe that early episode termination, e.g., due to safety violations, is a powerful and ubiquitous signal available in various RL tasks. It is especially prevalent in robotic tasks such as autonomous driving and robot locomotion, where eventual real-world deployment is the end goal, and the safety of the agent and its surroundings is paramount.

---

**Algorithm 1** Updating Auxiliary Start State Distribution via Episode Length (*AuxSS*)

---

1: **Inputs:** Sampling distribution $\mathcal{W}$, Sampling distribution norm $\mathcal{N}$, Episode Length $L_{ep}$, Update index $i$, Task Horizon $H$, Offline Demonstration States $\mathcal{S}_{demo}$, Weight Threshold $\delta$, Smoothing Variance $\sigma^2$
2: **Outputs:** Sampling distribution $\mathcal{W}$, Sampling distribution norm $\mathcal{N}$
3: $\mathcal{W}[i] \leftarrow \text{MAX}(\frac{H - L_{ep}}{H}, \delta)$          ▷ $\delta$ ensures probability of sampling $\geq 0$
4: $\lambda \leftarrow \frac{1}{\sqrt{2\pi}\sigma}\text{EXP}\left(\frac{(S_{demo} - S_{demo}[i])^2}{2\sigma^2}\right)$      ▷ $\lambda$ is used for smoothing updates to $\mathcal{W}$.
5: $\mathcal{W} \leftarrow (1 - \lambda) * \mathcal{W} + \lambda \mathcal{W}[i]$     ▷ $*$ is an elementwise multiplication b/w arrays $\lambda$ and $\mathcal{W}$
6: $\mathcal{N} \leftarrow \text{SUM}(\mathcal{W})$

---

Intuitively, for a state $s$, if the proportion of actions that cause the agent to land in a terminal state is high, then a larger exploratory budget is required to learn a feasible action for this state by the policy. Moreover, the chance of navigating through this state likely hinges on the repeated selection of a small set of safe and feasible actions in the neighbouring regions of the state space. Therefore, there is a high likelihood that such states belong to $\mathcal{C}$ and sampling them more frequently can accelerate learning. We define $\Omega_\pi(s)$ as a notion of safety capturing one minus the probability of a policy $\pi$ causing early episode termination from a state $s$ after a $k$ step rollout. More formally, for any state $s \in \mathcal{S}$ as -

$$\Omega_\pi(s) = \int_{a_{0:k-1}} P(a_{0:k-1}|s,\pi) \int_{s_k} P(s_k|s, a_{0:k-1}, \mathcal{T}, \pi)\mathcal{Z}(s_k)\, ds_k \, da_{0:k-1} \tag{2}$$

Here, $\mathcal{Z}(s) \in \{0,1\} \, \forall s \in \mathcal{S}$ and denotes whether or not state $s$ causes episode termination. $\mathcal{Z}(s) = 0$ if episode termination is caused by being in state $s$ and $1$ otherwise. $a_{0:k-1}$ is the sequence of $k$ actions induced by the policy $\pi$ from state $s$ under the transition model given by $\mathcal{T}$. $s_k$ is the state that is reached when policy $\pi$ takes action sequence $a_{0:k-1}$ starting from state $s$ in an environment with transition model $\mathcal{T}$.

Exactly computing $\Omega_\pi(s)$ is still computationally expensive. To address this, we leverage the time to termination or episode length from a given start state, which is a freely available metric at training time, as a Monte Carlo approximation of the true state safety for a given policy at that state. By maintaining a parameterized distribution over a set of desirable start states, referred to as *candidate task-critical states* ($\widetilde{\mathcal{C}}$), we can exploit local smoothness in the majority of the state space to

quickly propagate these approximations across the start state distribution. These steps are described in Algorithm 1.

We now describe how we incorporate the expert demonstration data into training. Since this data comprises successful demonstrations of the task, the demonstration trajectories will likely contain task-critical states. Motivated by this observation, we set $\widetilde{\mathcal{C}}$ to be the states from the demonstration data and identify $\mathcal{C}$ from amongst these states over the course of training. Putting everything together, we get our proposed method *AuxSS*, illustrated in Figure 2 and described in Algorithm 2.

---

**Algorithm 2** Online RL with Auxiliary Start States

---

1: **Inputs:** Task Horizon $H$, Offline Demonstration States $\mathcal{S}_{demo}$, Algorithm $\mathcal{A}$, Training Timesteps $T_{max}$, Environment $\mathcal{E}$, replay buffer $\mathcal{B}$
2: Sampling distribution $\mathcal{W} \leftarrow \overbrace{[1, 1 \ldots 1]}^{\text{len}(\mathcal{S}_{demo})}$ $\quad \triangleright$ Initialization incentivizes visiting states atleast once
3: Sampling distribution norm $\mathcal{N} \leftarrow \text{SUM}(\mathcal{W})$
4: $t \leftarrow 0$
5: **while** $t \leq T_{max}$ **do**
6: $\quad i \leftarrow \text{SAMPLESTARTSTATE}(\frac{\mathcal{W}}{\mathcal{N}})$
7: $\quad s_0 \leftarrow \mathcal{S}_{demo}[i]$
8: $\quad L_{ep} \leftarrow \text{TRAINFORONEEPISODE}(\mathcal{A}, \mathcal{B}, \mathcal{E}, s_0))$ $\quad\quad \triangleright$ Return value is episode length
9: $\quad t \leftarrow t + L_{ep}$
10: $\quad \mathcal{W}, \mathcal{N} \leftarrow \text{UPDATESAMPLER}(\mathcal{W}, \mathcal{N}, L_{ep}, i, H, \mathcal{S}_{demo})$ $\quad\quad \triangleright$ See Algorithm 1

---

## 5 EXPERIMENTS

**Overview:** We first demonstrate state-of-the-art sample-efficiency and robustness of *AuxSS* on continuous sparse-reward hard-exploration tasks - one low dimensional 2D maze environment and two variants of a high dimensional 3D Navigation task in Miniworld Chevalier-Boisvert et al. (2023) with an image state space. Subsequently, we use a dense reward, easy exploration problem to show that in the absence of strong safety cues, the presence of task critical states $\mathcal{C}$ enables *AuxSS* to match algorithms with access to greater affordances. We then show that in hard exploration problems *AuxSS* is better suited for assimilating information from limited amounts of expert offline data by demonstrating better sample-efficiency with $15\times$ less offline expert data available to it. Finally, we empirically demonstrate that approximating a more uniform visitation distribution over $\mathcal{C}$ through $\Omega$ facilitates accelerated learning. We showcase how *AuxSS* is a good way to approximate $\Omega$ while other distributions not motivated by state safety are not.

**Setup:** We conduct our experiments on three testbeds - a low dimensional sparse reward continuous maze, two sparse reward high dimensional continuous 3D navigation tasks with image observations (the first is an easier exploration instantiation and the second a harder exploration variant) and a suite of three continuous control tasks in MuJoCo (Todorov et al., 2012) having dense rewards. For each task we assume access to one or two trajectories of demonstration data. For the maze tasks (referred to as Lava Bridge) this is 500 transitions, while for MuJoCo this is 1000. The mazes constitute hard exploration problems. They consist of large sections of untraversable regions that cause immediate termination along with a large negative reward if entered. The agent only gets a non-zero reward on reaching the goal state or entering a terminal state. More details are presented in Appendix 7.1. The MuJoCo task includes three environments - Ant-v4, HalfCheetah-v4 and Walker2D-v4. It constitutes an easier exploration problem where algorithms can quickly learn to avoid early episode termination (see Appendix 7.9).

### 5.1 DOES AUXSS ACCELERATE LEARNING OF ROBUST POLICIES?

In this section we study the efficacy of *AuxSS* at improving the sample efficiency of online learning by making use of affordances such as arbitrary resetting of the environment and access to a limited quantity of expert demonstration data by comparing *AuxSS* with online and hybrid methods. In addition to tracking sample-efficiency we also track the robustness of the learnt policy. The choice

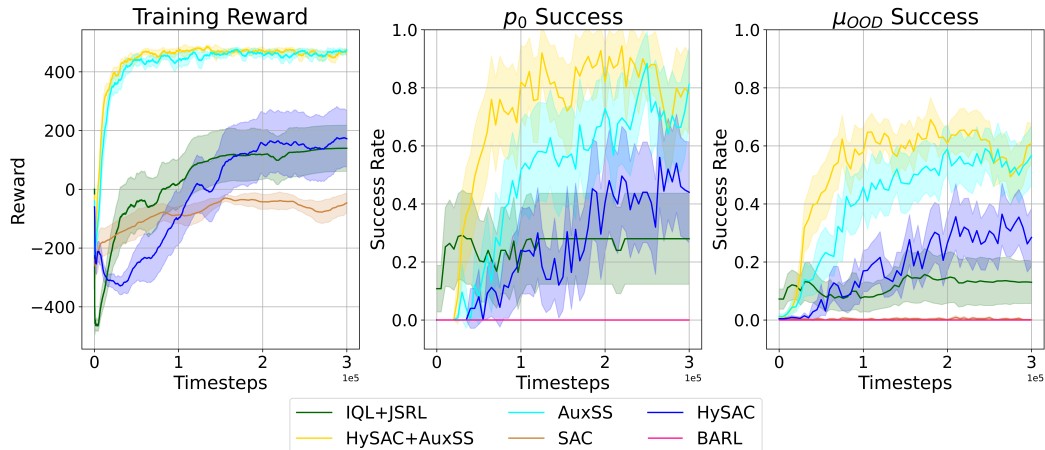

Figure 3: Task Completion Rate of Various methods on the Lava Bridge Environment. Each method is evaluated on an In Distribution (ID) and Out-of-Distribution (OOD) benchmark of starting states where the ID start state distribution is the start state distribution of the MDP while the OOD benchmark comprises a different distribution of start states.

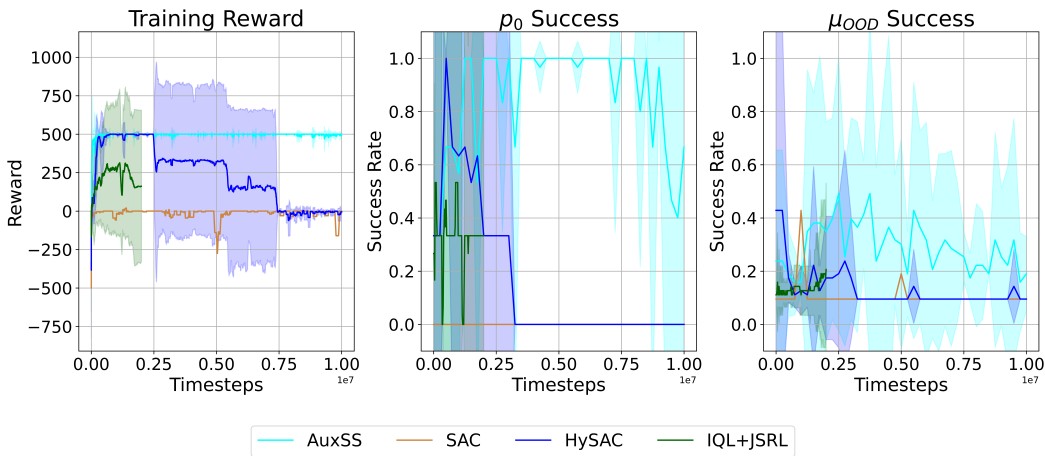

Figure 4: Task completion rate and training reward on the easy exploration instantiation of the 3D Navigation task.

of the MDP start state distribution ($p_0$) and robustness benchmark start state distribution ($\mu_{OOD}$), are shown in Figure 7.

Figure 3 presents the findings of this study on the Lava Bridge environment. A summary of the compared methods and the affordances these make use of are provided in Appendix 7.4 (see Fig 10). A standardized training setup has been used across methods (except BARL (Mehta et al., 2022)) where the number of offline demonstration transitions is set to 10 million, number of online learning steps is 300000, replay buffer size is 10000, max episode length is 500 and experiments are evaluated across 25 seeds. All hybrid methods have access to 500 transitions of expert demonstration data.

We use the example of SAC (Haarnoja et al., 2018), a purely online RL method, to highlight the exploration challenges that the Lava Bridge environment poses to standard online RL. SAC uses undirected entropy-based bonuses to promote exploration but struggles to efficiently explore in the Lava Bridge environment. Its failure to robustly reach the goal within the stipulated training budget highlights the exploration challenges posed by the Lava Bridge environment. In addition, we evaluate BARL (Mehta et al., 2022), an information-theoretic method for sample-efficient online exploration. We evaluated BARL both with and without access to the demonstration data. It failed

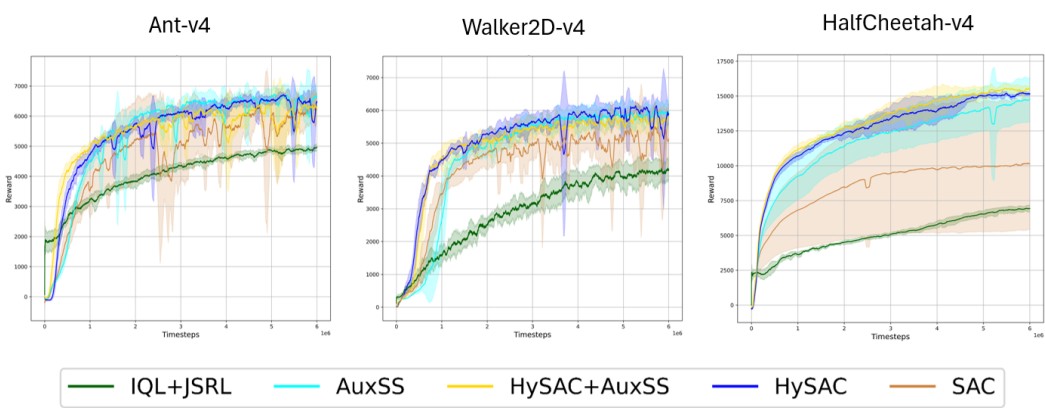

Figure 5: Reward on three continuous control tasks in MuJoCo.

to solve the Lava Bridge task in both cases. This is because BARL is reliant on a classical planner, which is designed to work with dense rewards. This is detrimental to performance in the Lava Bridge environment, which has a sparse reward, preventing the BARL planner from finding solutions in a limited time.

We compare our approach to two hybrid RL approaches - HySAC, an adaptation of HyQ (Song et al., 2023) where a DQN is replaced with SAC, and JSRL (Uchendu et al., 2023). Moreover, since our approach complements the persistent storage of offline demonstrations in HySAC, we also experiment with a combination of both (HySAC+*AuxSS*). The results can be seen in Figs. 3 and 6. Our approach is the most sample-efficient in terms of reward and success rates. Furthermore, combining our approach with HySAC yields better robustness in fewer training steps. By contrast, both HySAC and JSRL struggle to make full use of the limited offline demonstration data.

It can be noted that the approach taken by JSRL of handing over episode rollout from a guide policy to the learning policy is conceptually similar to having an auxiliary start state distribution that monotonically recedes towards $p_0$ over the course of training. Unlike our proposed auxiliary distribution, JSRL cannot reemphasize visiting previously learnt regions of the state space that may have been forgotten over the course of training. We have accounted for sample-efficiency gains that JSRL may obtain by directly resetting to the handover point (rather than using the guide policy to get there) by only tracking rollout steps beyond the handover point. Despite this, JSRL's inability to reemphasize visitation of previously learnt regions prevents it from learning very robust policies as can be seen in Figure 3.

### 5.2 DOES AUXSS SCALE TO HIGHER DIMENSIONS?

We evaluate AuxSS and other hybrid RL approaches on two 3D navigation tasks with image observation spaces to determine the how these methods fare in high dimensions (see Appendix 7.1.2) for details). Figure 4 presents results on the easier exploration version of the task while Figure 11 (see Appendix 7.5) presents results on the harder exploration variant. We find that AuxSS is the only method to consistently solve the easier instantiation while it is the only method to solve the harder instantiation from the original start state distribution.

### 5.3 ARE SAFETY INSPIRED AUXILIARY START STATES USEFUL FOR TASKS WITH LIMITED SAFETY CUES?

We investigate this question using the MuJoCo suite of tasks where early episode terminations cease rapidly, resulting in *AuxSS* becoming a uniform sampling distribution over expert states. Figure 5 shows the performance all methods on these MuJoCo tasks (averaged across 5 seeds). It can be noted that despite the absence of safety cues, *AuxSS* performs comparably to HySAC (Song et al., 2023) (while outperforming SAC (Haarnoja et al., 2018) and JSRL (Uchendu et al., 2023)) despite not having access to expert action and reward. We believe that this is due to the presence of task critical states $\mathcal{C}$ within the expert data that would have an even lower likelihood under the visitation

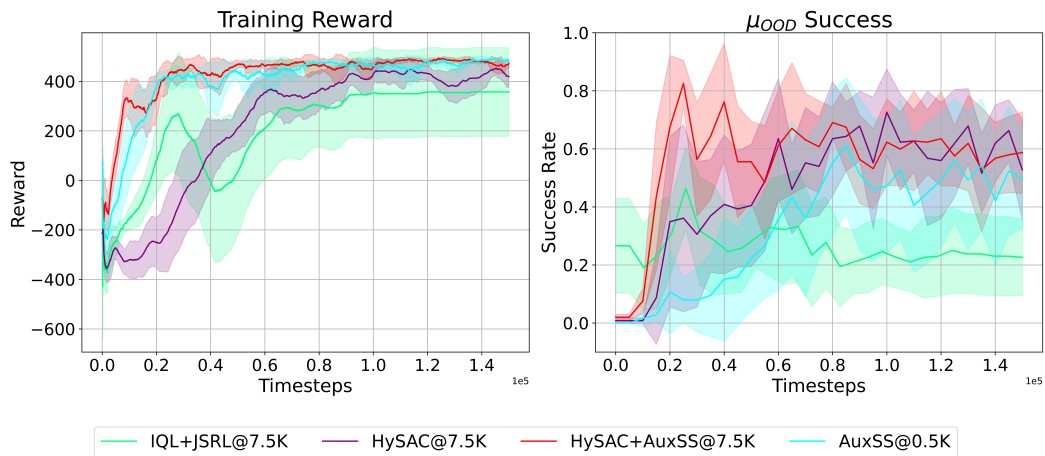

Figure 6: A study of how sample-efficiency and robustness vary for hybrid RL methods when provided with different amounts of demonstration data.

distribution of the original MDP start state as compared to the near uniform sampling distribution of *AuxSS*.

### 5.4 INFLUENCE OF OFFLINE DEMONSTRATION SET SIZE ON PERFORMANCE AND SAMPLE-EFFICIENCY

In Figure 6 we plot the training reward and evaluate robustness on the hard exploration task, Lava Bridge, when different quantities of expert demonstration data ($0.5K$ and $7.5K$ expert samples) are available prior to the online learning phase. We find that by accessing $15\times$ fewer expert samples *AuxSS* can match and exceed the robustness and sample efficiency of policies learnt via other hybrid RL methods. When provided access to a resetable simulator, this demonstrates that a good auxiliary start state distribution can more effectively assimilate data to guide exploration and accelerate learning than other approaches to hybrid RL. Unlike other methods, having a good start state distribution prevents the need to collect large quantities of expert data through ability to bootstrap online learning off of very limited demonstration trajectories

### 5.5 STATE SAFETY INSPIRED START STATE SAMPLING FOR SAMPLE EFFICIENCY

In Section 4, we connect the notion of state safety $\Omega$ with *task critical states* ($\mathcal{C}$) and discuss how this can influence sample-efficiency. In this section, we empirically validate our claims. We modify *AuxSS* by constructing a static distribution ($\Omega$-SS) that samples start states with respect to a random policy. Concretely, we sample start states inversely proportional to $\Omega_{\pi_{rand}}(s)$ where $\pi_{rand}(.|s) \sim \mathcal{U}^{|\mathcal{A}|}$. In practice, we use Monte Carlo sampling of actions for a fixed time horizon ($= 4$ time steps) to approximate this quantity for each state. Since the policy at the start of online training is initialized randomly, this mimics the state safety distribution with respect to the policy at the start of training. Therefore if our claims hold we expect to see matching sample-efficiency trends to *AuxSS* in the early stages of training.

Figure 7 presents the findings of this study. We see that as expected, $\Omega$-SS demonstrates matching sample-efficiency and robustness trends as *AuxSS* early in training. In fact, since $\Omega$-SS is the correct state safety distribution with respect to the initialized policy from the start of training it learns even faster than *AuxSS*, since *AuxSS* must gradually approximate this state safety distribution over the course of multiple training episodes.

We note here the divergence in robustness trends seen later in training. This is caused by the static nature of $\Omega$-SS which fails to adapt to the morphing $\mathcal{C}$ induced by the policy as it trains. This causes the resulting loss of robustness. The dynamic nature of *AuxSS* helps prevent this degradation as its able to adapt its start state distribution based on changes in the policy.

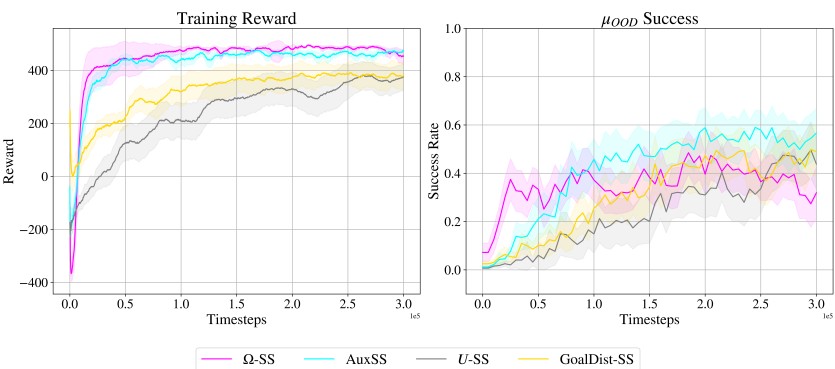

Figure 7: Sample-efficiency and robustness trends when simulator resets are selected using different start state distributions.

### 5.6 Do start state distributions not deriving from state safety fail to be sample efficient?

To study this inverse logical question, we construct two start state distributions, $\mathcal{U}$-SS and *GoalDist*-SS, that do not try to incentivize visitation of *task critical states*. $\mathcal{U}$-SS is a static distribution that uniformly samples states from the provided demonstrations. *GoalDist*-SS is a dynamic distribution that exponentially weights states based on their distance from the task goal. States closer to the goal are assigned a higher probability to be sampled. The time varying component of this distribution arises from temperature scaling of the distribution with the temperature gradually rising over the course of training. This promotes sampling near goal states early on in training and sampling more uniformly from the demonstration data later on in training. More details about these distributions can be found in Appendix 7.3.

Figure 7 contains the findings of this study. It can be seen that both $\mathcal{U}$-SS and *GoalDist*-SS are far slower to train than state safety inspired distribution demonstrating that not all start state distributions will accelerate learning. As a consequence of the poor choice of their state visitation, these methods fail to learn good policies in the stipulated training budget and thus also have much lower robust performance than *AuxSS* and $\Omega$-SS (before its static nature causes robustness to degrade).

## 6 Discussion and Limitations

In this work, we explore the use of commonly available affordances in RL tasks to guide online exploration. We highlight the importance of auxiliary start state distributions, constructed by utilizing small quantities of expert demonstration comprising only state information, in facilitating sample-efficient learning of robust policies. We find that in environments that allow arbitrary state resetting, this is a very crucial design choice and we observe that deriving start state distributions from notions of state safety can dramatically accelerate policy learning online. In terms of the Go-Explore philosophy of disentangling the choice of exploration frontier and how to get there, this work sheds new light on how the choice of exploration frontier can greatly influence sample-efficiency particularly in hard exploration tasks. While the need for a simulator that supports arbitrary state resets can be viewed as limiting, it is important to observe that most notable breakthroughs in RL have come on the back of powerful simulators that enable such resetting. Consequently, understanding how to effectively utilize such an affordance is a pertinent question that this work seeks to address.

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

# 7 APPENDIX

## 7.1 ENVIRONMENTS

In this section we describe observation, action and reward for the two families of maze environments used in our experiments.

### 7.1.1 LAVA BRIDGE

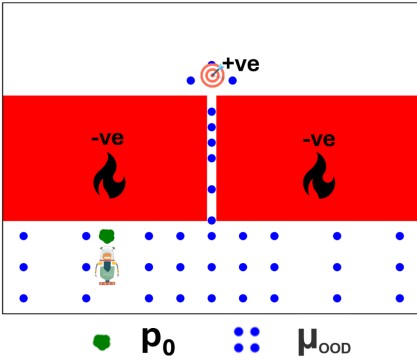

Figure 8: An illustration of the Lava Bridge environment. The red regions are the lava pits, the green blobs denote the MDP's initial start state distribution, $p_0$, and the blue spots correspond to the distribution start state distribution $\mu_{OOD}$. The red target marks the goal location.

**Observation Space:** We use a $4D$ state comprising position and velocity.

**Action Space:** A continuous $2D$ action space $[acc_x, acc_y]$ is employed. It comprises linear acceleration along the two axes. The agent has a mass (1 kg in our experiments).

**Reward Function:** The agent receives $+500$ for reaching the goal, $-500$ and immediate episode termination for touching the lava pits and $0$ otherwise.

### 7.1.2 3D NAVIGATION

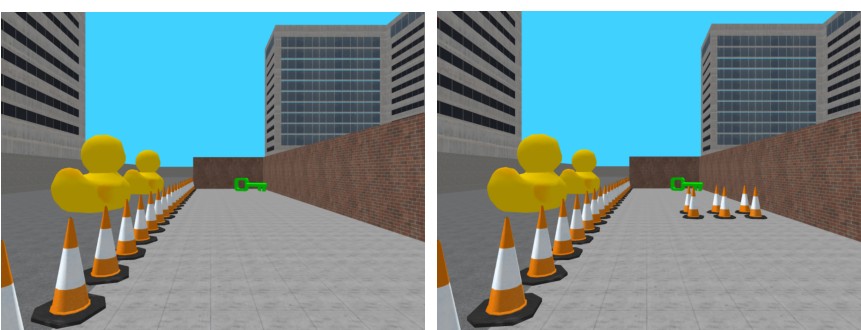

Figure 9: (*Left*) *Easy* instantiation (*Right*) *Hard* instantiation of the 3D navigation task.

**Observation Space:** We use images as the observation space for this family of tasks. Specifically all policies receive a down-sampled first person view of the scene from the agent perspective as a $32x32$ greyscale image. This is flattened and appended with the heading angle (a continuous $1D$ value) and velocity (a continuous $2D$ value) to form a continuous 1027 dimensional observation space.

**Action Space:** A continuous $2D$ action space $[a_{acc}, a_\omega]$ is employed. Here, $a_{acc}$ is the linear acceleration along the heading direction and $a_\omega$ is the angular velocity. The agent has a mass (10 kg in our experiments).

**Reward Function:** The agent receives $+500$ for reaching the green key, $-500$ and immediate episode termination for touching an orange cone and $0$ otherwise

## 7.2 PRELIMINARIES

We track the performance of our policies on a set of start states $\mu_{OOD}$ that are out-of-distribution with respect the start state distribution $p_0$. The object under this setup can be formalized through the following expression:

$$\mathcal{J}_{\mu_{OOD}}(\pi) = \mathbb{E}_{s_0 \sim \mu_{OOD}, s_{t+1} \sim \mathcal{T}(s_t, a_t), a_t \sim \pi(s_t)} [\Sigma_{t=0}^{H} \gamma^t r(s_t, a_t, s_{t+1})] \tag{3}$$

## 7.3 START STATE DISTRIBUTIONS

In this sections we concretely describe the other start state distributions that *AuxSS* is compared with.

The $\Omega$-SS is the correct state safety distribution with respect to the initialized random policy at the start of training. It samples states based on state un-safety where the probability of sampling a start state is based on the precomputed state un-safety value as given by

$$\Omega_\pi(s) = \int_{a_{0:k-1}} P(a_{0:k-1}|s, \pi) \int_{s_k} P(s_k|s, a_{0:k-1}, \mathcal{T}, \pi) \mathcal{Z}(s_k) \, ds_k \, da_{0:k-1} \tag{4}$$

where, $\mathcal{Z}(s) \in \{0, 1\} \, \forall s \in \mathcal{S}$ and denotes whether or not state $s$ causes episode termination. $\mathcal{Z}(s) = 1$ if episode termination is caused by being in state $s$ and $0$ otherwise. Note how this is the same as Equation 2 with subtle difference that $\mathcal{Z}(s) = 1$ now refers to episode termination causing states.

*GoalDist*-SS incentivizes visitation of *task critical states*. The probability $p$ of sampling $i^{th}$ from demo data $S_{demo}$ with $N$ demo states and given goal state $g$ is

$$p(S_{demo}[i]) = \frac{e^{\frac{(S_{demo}[i]-g)^2}{\tau}}}{\Sigma_j^N e^{\frac{(S_{demo}[j]-g)^2}{\tau}}} \tag{5}$$

Here $\tau$ is a time varying temperature scaling coefficient which makes the distribution go from peaked to uniform over the course of training. It varies inversely to the cosine of the fraction of training complete. Towards the end of training this causes the sampling distribution to be nearly uniform.

$\mathcal{U}$-SS is simply a uniform distribution over the set of start states available.

## 7.4 AN AFFORDANCE BASED COMPARISON OF RL APPROACHES

Figure 10 presents a comparison of various approaches based on the affordances leveraged by the methods.

| Method | Affordance | | | |
|---|---|---|---|---|
| | 👁 | 🎁 | 🗡 | 📍 |
| SAC | ✗ | ✗ | ✗ | ✗ |
| BARL | ✓ | ✓ | ✓ | ✓ |
| HySAC | ✓ | ✓ | ✓ | ✗ |
| JSRL | ✓ | ✓ | ✓ | ✓✗ |
| AuxSS *(Ours)* | ✓ | ✗ | ✗ | ✓ |
| HySAC + AuxSS *(Ours)* | ✓ | ✓ | ✓ | ✓ |

👁 State    🎁 Reward    🗡 Action    📍 Resets

Figure 10: This table contrasts the compared methods in terms of the affordances they make use.

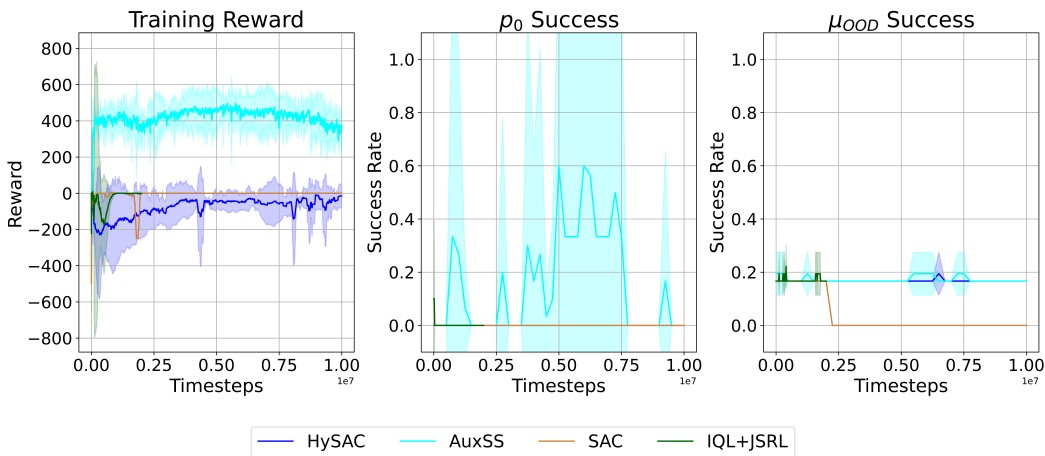

Figure 11: Task completion rate and training reward on the hard exploration instantiation of the 3D Navigation task.

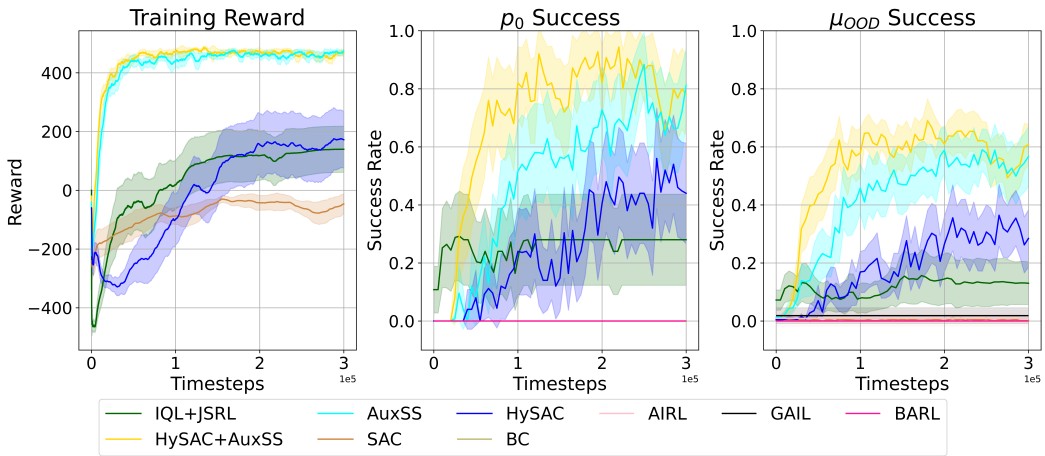

Figure 12: Extended evaluation with imitation learning algorithms on the Lava Bridge Environment. Each method is evaluated on an In Distribution (ID) and Out-of-Distribution (OOD) benchmark of starting states where the ID start state distribution is the start state distribution of the MDP while the OOD benchmark comprises a different distribution of start states.

### 7.5 PERFORMANCE ON THE HARD INSTANTIATION OF THE 3D NAVIGATION TASK

Figure 11 presents results on the hard exploration variant of the 3D navigation task. AuxSS is the only method that is able to solve the task from the original start state distribution.

### 7.6 COMPARISON WITH IMITATION LEARNING METHODS ON LAVA BRIDGE ENVIRONMENT

We present additional comparisons with imitation learning methods on the lava bridge environment. We compare with behavior cloning, GAIL Ho & Ermon (2016) and an inverse RL method, AIRL Fu et al. (2018). From Figure 12 we can see that imitation learning methods fail at this task and cannot compete with any hybrid RL approach.

### 7.7 HOW DOES NUMBER OF DEMONSTRATIONS IMPACT AUXSS?

Figure 13 presents an ablation over various task horizons on the Easy instantiation of the 3D navigation task. Since AuxSS simply utilizes demonstrations for picking resetting candidates only, we find

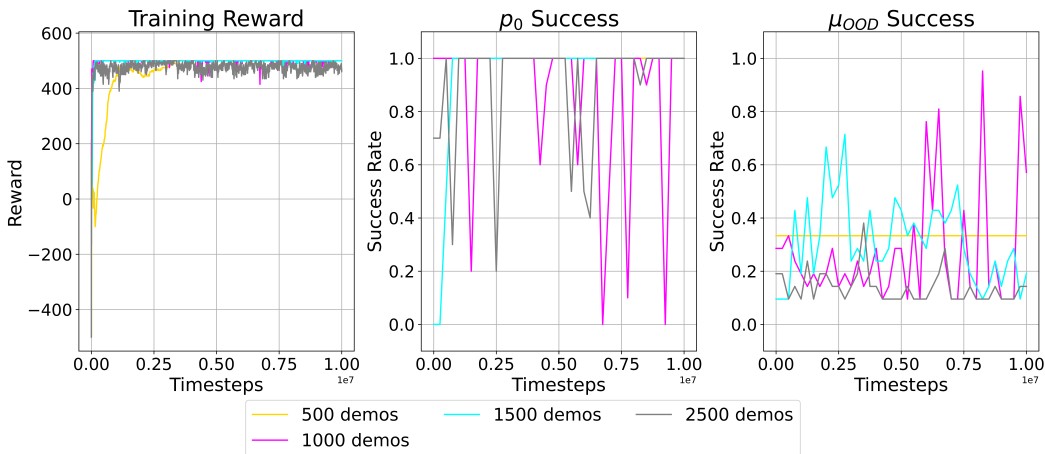

Figure 13: Performance on the *Easy* instantiation of the 3D navigation task in Miniworld when varying the number of samples present in the offline demonstration data.

that it isn't too sensitive to the demonstration set size. Instead we believe that the algorithm is more dependent on the quality of the expert demonstration. In addition the diversity of the demonstrations can influence the robustness of the learnt policies as seen by the variance in the robust benchmark.

## 7.8 HOW DOES THE TASK HORIZON IMPACT AuxSS?

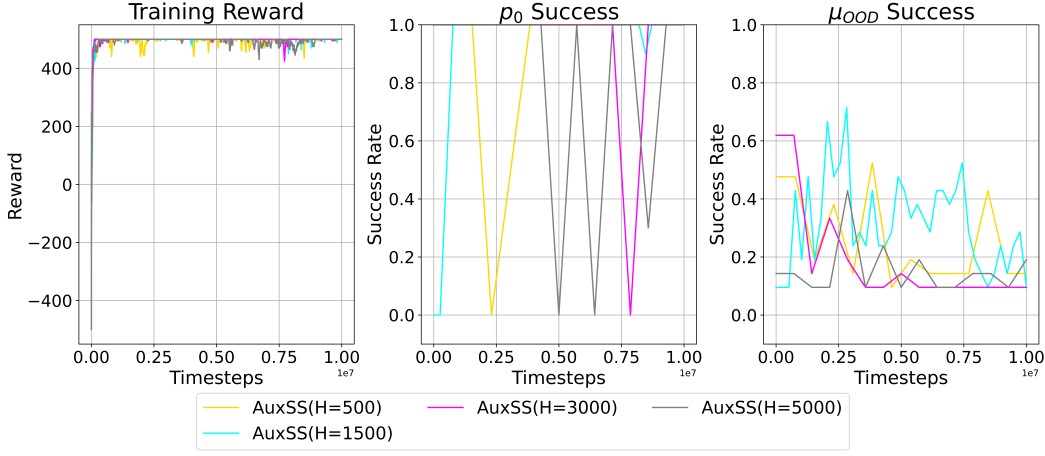

Figure 14: Performance on the *Easy* instantiation of the 3D navigation task in Miniworld when varying the task horizon (described by $H$ and used in Algorithm 1

Figure 14 presents an ablation over various task horizons on the Easy instantiation of the 3D navigation task. We find that the method is able to successfully complete the task across different task horizons. We see some regression in robustness at very small and large task horizons. This could be explained by an alteration in the visitation distribution induced by the sampling strategy and the time the agent gets to spend exploring from a certain state. While we ourselves haven't tuned parameters within the algorithm, we believe that tuning the smoothing variance to ensure propagation of termination signal across the sampling distribution should be able to help improve performance no matter the task horizon.

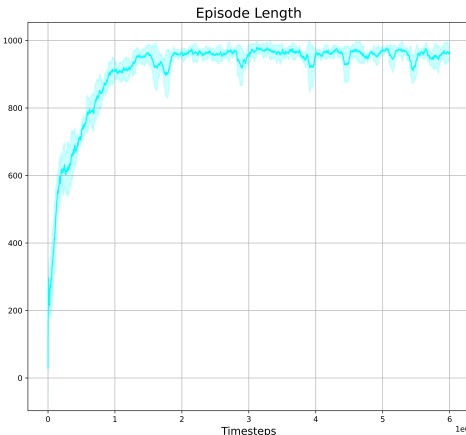

Figure 15: Rapid saturation of episode length based signal in MuJoCo.

## 7.9 RAPID SATURATION OF EPISODE LENGTH IN MUJOCO

In MuJoCo tasks where early episode termination is observed, we notice a rapid saturation of episode length (see Figure 15) which we attribute to the easy exploration problem facilitated by the dense reward feedback in these tasks.

## 7.10 HOW DOES THE SAMPLING DISTRIBUTION LOOK OVER TRAINING?

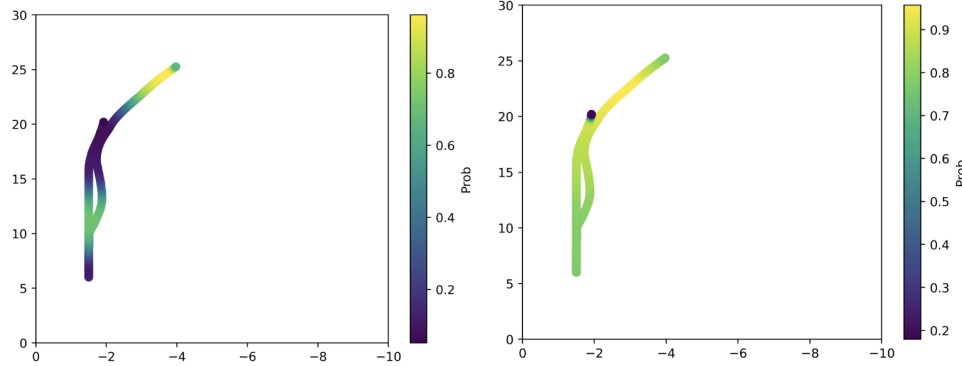

Figure 16: (*Left*) Sampling distribution over start states early on in training on the Hard instantiation of the 3D Navigation task. (*Right*) Sampling distribution over start states late on in training on the Hard instantiation of the 3D Navigation task.

Figure 16 illustrates how the sampling distribution changes from a termination centric distribution at the start of training to a uniform distribution at the end of training when good behavior has been learned everywhere.

