# OpenReview forum: "Accelerated Online Reinforcement Learning using Auxiliary Start State Distributions"
_ICLR.cc/2025/Conference — Submitted to ICLR 2025_

### Official Review · Reviewer_1Utq · 2024-11-04

**Soundness:** 2
**Presentation:** 3
**Contribution:** 2
**Rating:** 5
**Confidence:** 3

**Summary:**

This paper addresses the issue of sample efficiency in online reinforcement learning (RL). The authors propose a method, Auxiliary Start State Sampling (AuxSS), which leverages a small set of expert demonstrations and a simulator with arbitrary resets to improve exploration and policy robustness. AuxSS uses an auxiliary start state distribution informed by safety cues—essentially prioritizing task-critical states where safety violations, such as early episode terminations, commonly occur. This approach enables better exploration in environments with sparse rewards and difficult exploration tasks, as it targets states crucial for task completion.

**Strengths:**

Overall, the paper is well-written and easy to follow. The method is presented clearly with sufficient notations. The idea is interesting and straightforward. The algorithm is compatible with many RL algorithms and can be applied to many scenarios.

**Weaknesses:**

The below paper seems to be highly relevant, but the authors didn't discuss and compare with it:
- Contrastive Initial State Buffer for Reinforcement Learning (https://arxiv.org/abs/2309.09752v3), which comes with open-sourced code (https://github.com/uzh-rpg/cl_initial_buffer).

Besides, the current experiments only covered a 2D discrete env (lava bridge) and 3 Mujoco task. How would the algorithm perform on high-dimensional tasks with sparse rewards? (the hard tasks in MetaWorld for example)

**Questions:**

1. Is the algorithm applicable to environments with more randomness? For example, a maze task where the start location, the goal, and the map are all randomly generated for each episode. What would be "task-critical states" for such scenarios?
2. In terms of "the time to termination"(line 252), do you only consider failure episodes or both successful and failure episodes? From line 240, it should consider the cases of "early episode termination", while it doesn't distinguish between successful and failed runs in algorithm 1.
3. How is task horizon H determined and how does it influence the algorithm performance? Does the algorithm work for dynamic-horizon tasks?
4. In Figure 7, it would be better to include the baseline algorithms' performance with 0.5K demonstrations as well. Currently, we don't know if the performance of the baseline algorithms drops with fewer demonstrations.

---

> ### Author Response · Authors · 2024-12-03
> **Thank you for your time and constructive feedback!**
>
> ## **Besides, the current experiments only covered a 2D discrete env (lava bridge) … algorithm performed on high-dimensional tasks with sparse rewards?**
>
> Thank you for your feedback! We have included a new high dimensional sparse reward 3D navigation environment using the Miniworld simulator where we learn an **image-based policy** directly mapping pixels to actions. Two versions with different hardness of exploration have been instantiated. To summarize, we showcase that in these hard exploration high dimensional continuous control tasks where the goal is to learn a policy from images, AuxSS is the best performing method and more importantly is the only method that can solve the given task when faced with hard exploratory challenges. These new results can be found in Section 5.2  and Appendix 7.5 of the updated draft..
>
> In addition, we would like to apologize for the confusion and clarify that the lava bridge is a continuous state and continuous action environment. We will make this clearer in the final version of the paper.
>
> ## **The below paper [1] seems to be highly relevant, but the authors didn't discuss and compare with it**
>
> We thank the reviewer for sharing. We would however like to highlight two factors that make this method orthogonal to the proposed approach.
>
> - [1] creates a sampling distribution based on the change in value function in a  **dense reward** setup. The reliance on dense reward feedback is incompatible with the **sparse reward, hard exploration** setups studied by us, where the predominant reward is $0$ and task relevant reward that guides the policy to improve on the task can often be never seen  (based on the choice of algorithm). As such [1]  will have insufficient information to compute meaningful advantage estimates that are necessary for its operationalization. Consequently, [1] is geared more towards a dense reward setup while AuxSS is designed to accelerate sparse reward, hard exploration problem making the two methods orthogonal to each other
>
> - [1] has only been shown to work on very low dimensional problems (under <10 dimensions). It requires contrastively learning state embeddings continually alongside the policy. This doesn’t trivially scale to images. In our experiments, the 3D Navigation task with its high dimensional visual observation space poses a computational and engineering challenge to get [1] working. AuxSS on the hand scales to image observation spaces trivially.
>
> Moreover it is operationalized via an advantage function that takes expert action which we hope to avoid using in order to construct the sampling distribution of states.
>
> Nonetheless we would like to thank the review for sharing [1]. We will include a discussion on it in the related works of the final draft of the paper.
>
> Additionally we would like to highlight that we have included three new imitation learning baselines that include BC, AIRL and GAIL on the Lava Bridge environment. We find that even on this relatively simpler lower dimensional problem these methods fail to learn a policy that can solve the task. Results for this can be found in Appendix 7.6 and Figure 12.
>
> [1] Nico Messikommer, Yunlong Song, Davide Scaramuzza  "Contrastive Initial State Buffer for Reinforcement Learning"
>
> ## **In terms of "the time to termination"(line 252), do you only consider failure episodes or both successful and failure episodes? From line 240, it should consider the cases of "early episode termination", while it doesn't distinguish between successful and failed runs in algorithm 1.**
>
> That is a great question!  We do not distinguish between the two for the following reason:
>
> Bellman bootstrap. In sparse reward setups, considering time to termination for only failure episodes is insufficient to ensure that the value function around states leading to failure is informative and useful for bootstrapping. By treating successful episodes the same way as failures, we ensure that the value function updates that arise from successfully reaching the goal are spread across the state space. Simultaneously upweighting states that lead to episode failure enables these to be the first set of unsolved states to benefit from good bootstrap targets of the value function that have been propagated. In Appendix 7.10 we have added visualizations of the sampling distribution to showcase this by presenting how the algorithm incentivizes start states close to successful terminations (towards the upper end of the trajectory where the goal is located) and also in unsafe states (coinciding with the constricted passage between the two sets of cones - See Figure 9 (right) ) in the early stages of learning.

---

> > ### Author Response · Authors · 2024-12-03
> >
> > ## **Is the algorithm applicable to environments with more randomness? For example, a maze task where the start location, the goal, and the map are all randomly generated for each episode. What would be "task-critical states" for such scenarios?**
> >
> > This is a very interesting question! Since AuxSS is simply a sampling algorithm it will be generally applicable even in cases where randomness exists within the environment. The ability to deal with randomness would be determined by other aspects such as the choice of observation space, the representation of the state and whether the simulator can be reset from such a representation. To give you a concrete example, assume two instantiations of a 3D variant of the lava bridge environment (like Miniworld) having an image observation space. Even in different instantiations of the map, an agent facing a pool of lava will observe a very similar-looking image. If the representation of this observation is close enough and the simulator has the ability to reset from such a representation, we would expect AuxSS to be equally useful across maps. Both in the Lava Bridge environment and the 3D Navigation task we evaluate policies from a broader set of start states than the original start state distribution. Performance on this robust benchmark sheds some light on applicability  in environments with more randomness. The high dimensional 3D navigation task, where AuxSS continues to outperform baselines, in particular is informative in this regard as the robust benchmark entails starting from viewpoints that may have been relatively under visited during normal training. To conclude we agree that these are all very pertinent questions and considerations and believe that understanding the effect of more randomness deserves a more thorough analysis as a follow-up work.
> >
> > ## **In Figure 7, it would be better to include the baseline algorithms' performance with 0.5K demonstrations as well. Currently, we don't know if the performance of the baseline algorithms drops with fewer demonstrations.**
> >
> > Apologies for the confusion. To clarify, besides Figure 6 in the updated draft, the other figures containing results on Lava Bridge all contain baselines that are computed with 0.5K samples. In particular, Figure 3 contains the comparison of baseline algorithms having access to 0.5K samples with AuxSS when it has access to 0.5K samples. From there we can see that AuxSS outperforms baselines in the low data regime. To better convey this we will further improve the wording for the camera ready if accepted.
> >
> > # **How is task horizon H determined and how does it influence the algorithm performance? Does the algorithm work for dynamic-horizon tasks?**
> >
> > Thank you for bringing this up! There is no fixed guideline for determining the task horizon. As a rough rule of thumb we set H to be some multiple of the length of a successful demonstration trajectory. This provides the learning algorithm to explore sufficiently far while ensuring that the feasible solution will be possible within this limit set by H. To further examine this question, we have added an ablation over H in Appendix 7.8. So far, we have only been able to conduct a single experiment, but we will include more in the final version. The choice of H doesn’t impact performance when sampling the initial state from $p_0$. However, when the initial state is out of distribution, very large H values appear to deteriorate performance. However, we note that this corresponds to results obtained with a single seed, and we will run more seeds in the coming weeks and include those results should the paper be accepted.

---

> > > ### Author Response · Authors · 2024-12-03
> > > **Update to IQL+JSRL in Figure 4 and Figure 11**
> > >
> > > We would also like to share that we completed running IQL+JSRL (5 seeds) on both 3D navigation environments for 10 million steps and continue to see it underperform AuxSS. We will update these figures in the final version of the paper. Below we share the high level trends -
> > > - On the easy instantiation of the task, IQL+JSRL achieves a max $p_0$ success rate of $80$% vs $100$% achieved by AuxSS. Additionally, its out-of-distribution success rate remains less than $50$% of AuxSS
> > > - On the hard instantiation  of the task, IQL+JSRL also fails to solve the task from the initial start state distribution while AuxSS completes the task

---

> ### Author Response · Authors · 2024-12-04
> **Updated ablations with 3 seeds per experiment**
>
> We would like to share that we improved the trustworthiness of the ablations over the demonstration size (Fig 13) and episode horizon (Fig 14) by running each experiment with 3 seeds. The updated figures will be included the final version of the paper. We find that in both cases AuxSS is insensitive to changes in these parameters and the performance of AuxSS remains nearly the same.
>
> - For the ablation over demonstration size this makes sense since AuxSS is dependent on the completeness / quality of the episode rather than the quantity of demonstration transitions.
>
> - For the ablation over episode horizon, we believe that despite the increase in episode horizon, the episode horizon isn't big enough to flatten out the sampling distribution and it maintains enough "peakiness" in the early stages of training to incentives sampling start states near episode terminations.

---

### Official Review · Reviewer_e98q · 2024-11-04

**Soundness:** 2
**Presentation:** 2
**Contribution:** 2
**Rating:** 3
**Confidence:** 4

**Summary:**

This paper proposes AuxSS, aiming to improve sample efficiency in online RL by sampling the initial states from an auxiliary distribution. The distribution is dynamically updated during training using a Monte Carlo-like scheme. With such auxiliary start distribution, the policy can avoid struggling at the start region of the environment, thus enjoying higher sample efficiency and safety.

**Strengths:**

- The problem discussed in this paper is novel. Low sample efficiency is a long-existing challenge in online RL, and this paper provides a novel perspective to further address this problem.
- The experiments cover various settings, including both sparse- and dense-reward tasks and different common baselines, showing the effectiveness of the auxiliary start distribution.

**Weaknesses:**

See questions.

**Questions:**

- What is the formulation of $J_{\mu_{OOD}}$?
- I do not find any reference of Figure 2. Could you please insert it to the proper place to better explain your algorithm? Besides, I wonder what the color of points in the middle of Figure 2 represents.
- The notation in Algorithm 1 is not clear. For example
  - What does $S_{demo} - S_{demo}[i]$ means if $S_{demo}$ is a state sequence while $S_{demo}[i]$ is a single state?
  - Following the previous question, why $\lambda$ is computed by this equation?
  - Is $W[i]$ the $i$-th position of $W$, or a new variable? If it is a part of $W$, Line 3 has changed its value, so the update in Line 5 is meaningless. If it is a new variable, what does Line 5 do by adding a sequence and a single value?
- The description of Lava Bridge appears at the beginning of Section 5, but the illustration is at Figure 8, Section 5.4, which is quite confusing. Could you please move this illustration to the same place of the description?
- AuxSS is based on the early termination of episodes. However, HalfCheetah-v4 will not terminate until it reaches the timestep limitaion, that is, it will not early terminate. Including this environment does not consistent to the motivation of this paper. Could you please use environments with early termination, such as Hopper or Humanoid?
- Is the Lava Bridge environment first proposed in this paper? If it is, could you please provide detailed information about it, especially the reward function? You only mentioned that "The agent only gets a non-zero reward on reaching the goal state or entering a terminal state", but not giving how much reward the agent will get. As a result, I am confused by the relation between the training reward and the success rate.
- In Section 5.3, I think the performance change of AuxSS over the size of expert demonstration should be included. Will increasing the size of demonstration further promote the performance of AuxSS? And will AuxSS work if the demonstrations are expert but half-way trajectories?
- In Section 5.4, the sampling distributions are unclear. What are the specific formulations of $\Omega$-SS and GoalDist-SS?
- Could you please illustrate how $W$ changes during the training process?

---

> ### Author Response · Authors · 2024-11-29
> **Thank you for your time and constructive feedback**
>
> Dear reviewer, thank you for your valuable suggestions. We have made changes to address your concerns and strengthen the paper.
>
>
> ## **AuxSS is based on the early termination of episodes. However, HalfCheetah-v4 will not terminate until … this environment is not consistent with the motivation of this paper.**
>
> This is a great point! The motivation for including MuJoCo tasks in general and HalfCheetah-v4 specifically is to showcase the competence of the proposed approach in a setting that does not play to its strengths. These  experiments highlight that even in the absence (or rapid saturation)  of early episode termination, a uniform reset distribution over the start states along an expert demonstration is  beneficial since these states constitute a superset of important but unlikely states (referred to as task critical states in the paper) for the task. Sampling these states even with a uniform distribution results in better sample complexity than training from just the original start state distribution as seen in Figure 5 of the updated draft.  We will make this clearer in the final version of the paper.
>
> ## **Could you please use environments with early termination, such as Hopper or Humanoid?**
>
> Yes! We have included evaluations on a new 3D navigation task using the Miniworld simulator. This environment is far more high dimensional than Hopper and Humanoid, having an image observation. Using this we showcase that in high dimensional continuous control tasks, AuxSS is the best performing method and more importantly is the only method that can solve the given task when faced with hard exploratory challenges. These new results can be found in Section 5.2  and Appendix 7.5 of the updated draft. Note that JSRL is trained partway at the time of submission. It substantially lags behind AuxSS and we will update the plot with the complete run for the final version of the paper.
>
> ## **In Section 5.3, I think the performance change of AuxSS over the size of expert demonstration should be included. Will increasing the size of demonstration further promote the performance of AuxSS?**
>
> Thank you for this valuable suggestion! We have included an ablation of size of expert demonstrations in the Appendix 7.7. We find varying the size of expert demonstrations does not impact the speed or efficacy of solving the task from the initiation start state distribution p0. We do see some variations in the robust performance of the algorithm, however, this appears more to do with quality and diversity of demos present rather than quantity as we see an initialization with 2500 demos underperforming the experiment utilizing fewer demos. We recognize that this trend can be noisy since only 1 seed has been used due to the paucity of time and finite compute resources. We will ensure to run more seeds in the coming weeks and include those results should the paper be accepted.
>
>
> ## **And will AuxSS work if the demonstrations are expert but half-way trajectories?**
>
> Interesting question! As discussed above, AuxSS is more dependent on the quality and diversity of the expert demonstration than quantity. Therefore we expect the efficacy of AuxSS to diminish if only trajectories halfway to the goal are provided. Nonetheless we will continue to see AuxSS prioritize sampling unsafe states within this set of expert states and once the value function has learnt good values to bootstrap training from states on this half-way trajectory, we expect AuxSS to provide improvements from that point on in training.
>
> ## **What is the formulation of $J_{\mu_{OOD}}$**
>
> $J_{\mu_{OOD}}$ is the expected reward to go from a start state distribution $\mu_{OOD}$. We have included the precise equation of this reward function in Appendix 7.2.
>
> ## **I do not find any reference of Figure 2. Could you please insert it to the proper place to better explain your algorithm? Besides, I wonder what the color of points in the middle of Figure 2 represents.**
>
> Thank you for this suggestion! We have now referenced Figure 2 in Section 4. The color of the points in the figure depict the probability of sampling the start state. We have also updated the caption of the figure to explain this.
>
> ## **Is the Lava Bridge environment first proposed in this paper? If it is, could you please provide detailed information about it, especially the reward function? ...  I am confused by the relation between the training reward and the success rate.**
>
> Thank you for pointing this out. The Lava Bridge environment is inspired by a similar environment used in BARL [1], however, the physics and maze layout are new in this work. We have improved the details of our environments and added details regarding reward, observation and action space in Appendix 7.1
>
> [1] Viraj Mehta, Biswajit Paria, Jeff Schneider, Willie Neiswanger, and Stefano Ermon. An experimental design perspective on model-based reinforcement learning. In International Conference on Learning Representations, 2022

---

> ### Author Response · Authors · 2024-11-29
> **Rebuttal continued (part 2)**
>
> ## **In Section 5.4, the sampling distributions are unclear. What are the specific formulations of $\Omega$-SS and GoalDist-SS?**
>
> We apologize for the confusion. $\Omega$-SS is a safety distribution based on state safety defined in Section 4 and GoalDist-SS is based on distance of the sampling state from the goal. We have a added the exact expressions for these distributions in Appendix 7.3 of the updated draft
>
> ## **Could you please illustrate how $W$ changes during the training process?**
>
> Yes! We included details of how $W$ changes  during the course of the 3D navigation task in Appendix 7.10
>
> ## **Clarifications about Algorithm 1**
>
> Thank you for bringing this up! The purpose of Line 5 is to perform a weighted average update to  $\mathcal{W}$. Similar states are likely to have very similar safety characteristics. We instantiate this prior through $\lambda$, which is a smoothing term spreading the newly updated $\mathcal{W}[i]$ to nearby states. This smoothing is computed using line 4. We find that such smoothing is essential for rapid propagation of useful safety scores across the offline demonstrations and is key to the successful working of AuxSS. As a clarification, $S_{demo} - S_{demo}[i]$ is a standard broadcasting between an array and scalar which results in a new array where each element in $S_{demo}$ is subtracted by the value of $S_{demo}[i]$.

---

> > ### Comment · Reviewer_e98q · 2024-11-30
> >
> > Thank you for your detailed responses to my comments. However, I still have some concerns regarding your revised draft and your responses, as outlined below:
> >
> > - The formulation of the smoothing parameter $\lambda$ remains unclear. Intuitively, as a smoothing parameter, $\lambda$ should lie within the range $[0,1]$. However, the formulation in Line 4 of Algorithm 1 does not guarantee that $\lambda \in [0,1]$. Furthermore, when $S_{demo}[j]$ is far from $S_{demo}[i]$, there is a risk of overflow for $\lambda[j]$. How do you handle this potential issue?
> >
> > - The experiments in these sections were conducted using only one seed. Given the importance of experimental reproducibility and the potential for seed-dependent variability, I am unable to fully trust the conclusions drawn from these results at this time.
> >
> > - In Figure 15, only the change in episode length using AuxSS is shown, without any comparison to other baseline methods. Including comparisons to other approaches would strengthen the validity of your claims and provide clearer context for the performance of AuxSS.
> >
> > - It appears that the sum of the sampling probabilities in the right figure exceeds 1. However, since the sampling distribution $\mathcal{W}$ is discrete, the total sum should be exactly 1. Could you clarify what the probabilities in Figure 16 actually represent and address the discrepancy?
> >
> > In summary, while the paper presents a well-motivated approach and demonstrates strong performance across various tasks, the writing and experimental robustness require further refinement. At this stage, I am inclined to consider rejecting the paper, pending further clarifications and improvements.

---

> ### Author Response · Authors · 2024-11-30
>
> Thank you for these questions!
>
> - That is a great question! Since Line 4 is the pdf of a normal distribution with a fixed standard deviation, we simply pick a $\sigma$ that ensures that the max value of the pdf is $<1$. In our experiments, we set $\sigma$ to either $1$ or $0.2$. Both these values ensure that $\lambda \in [0,1]$. Also, the overflow of $\lambda[j]$ doesn't pose an issue to the algorithm. Infact, it can sometimes be a desirable effect. An overflowing $\lambda[j]$ will simply upweight the probability of sampling a distant demo state. This can actually be helpful as it incentivizes revisiting states previously assumed to be safe. Since the policy is constantly being updated during training, its induced state visitation distribution also changes. This may result in a different safety associated with a previously visited safe state under the updated policy. Revisiting states can ensure that our approximated state safety through episode length is infact up to date. Therefore, overflow of $\lambda[j]$ should be viewed as a desirable effect. The degree of this effect can be controlled by appropriately setting $\sigma$.
>
> - We are running more seeds and will aim to provide you an overview of these updated results in the comments section before the discussion period ends.
>
> - We thank you for this feedback. We will include this in the final version of the paper if it is accepted.
>
> - That is correct, the plotted figure contains the un-normalized probability scores from Line 5 in Algo 1. These scores are normalized before sampling happens. Figure 16 highlights the relative difference in sampling probabilities between different states along the expert trajectory as training progresses. This wouldn't change whether the scores were normalized or unnormalized. However, we thank you for your suggestion and apologize for the confusion. We will use normalized probabilities in the final version of the paper if accepted.

---

> ### Author Response · Authors · 2024-12-03
> **Update to IQL+JSRL in Figure 4 and Figure 11**
>
> We would like to share that we completed running IQL+JSRL (5 seeds) on both 3D navigation environments for 10 million steps and continue to see it underperform AuxSS. We will update these figures in the final version of the paper. Below we share the high level trends -
> - On the easy instantiation of the task, IQL+JSRL achieves a max $p_0$ success rate of $80$% vs $100$% achieved by AuxSS. Additionally, its out-of-distribution success rate remains less than $50$% of AuxSS
> - On the hard instantiation  of the task, IQL+JSRL also fails to solve the task from the initial start state distribution while AuxSS completes the task

---

> ### Author Response · Authors · 2024-12-04
> **Updated ablations with 3 seeds per experiment**
>
> We would like to share that we improved the trustworthiness of the ablations over the demonstration size (Fig 13) and episode horizon (Fig 14) by running each experiment with 3 seeds. The updated figures will be included the final version of the paper. We find that in both cases AuxSS is insensitive to changes in these parameters and the performance of AuxSS remains nearly the same.
>
> - For the ablation over demonstration size this makes sense since AuxSS is dependent on the completeness / quality of the episode rather than the quantity of demonstration transitions.
>
> - For the ablation over episode horizon, we believe that despite the increase in episode horizon, the episode horizon isn't big enough to flatten out the sampling distribution and it maintains enough "peakiness" in the early stages of training to incentives sampling start states near episode terminations.

---

### Official Review · Reviewer_SGVK · 2024-11-05

**Soundness:** 2
**Presentation:** 2
**Contribution:** 2
**Rating:** 3
**Confidence:** 3

**Summary:**

The authors propose an algorithm for selecting start state distributions other than the given start conditions. They introduce a mathematical definition of safety that represents the probability of a policy triggering an early episode termination. And two practical algorithms inspired by this mathematical object. The first algorithm produces a sampling distribution over given offline demonstrations using the length of the episode as a proxy for task success. The second algorithm uses this distribution to sample start states, simulating from the sampled state.

The algorithm is evaluated in the sparse Lava Bridge Environment against Inter-Quartile Learning + Jump Start Reinforcement Learning, an offline reinforcement learning algorithm combined with an exploratory policy learning algorithm and HySAC, a hybrid algorithm that learns online and utilizes offline demonstrations. They are able to beat the provided baselines.

The algorithm is further evaluated in three simple continuous control MuJoCo tasks, matching the performance of HySAC, and with fewer offline data than other algorithms in the Lava Bridge Environment.

Finally, they compare against sampling uniformly and with a different heuristic in the Lava Bridge Environment.

**Strengths:**

The introduction of the safety state distribution is quite interesting and the experiments in 5.4 and 5.5 provide useful insight into the benefits of this metric. Because of this, the algorithm is motivated in principle and is quite easy to implement.

**Weaknesses:**

Unfortunately the baselines in the main experiments are not designed for and do not have access to resetting to arbitrary states and are not directly comparable. JSRL seems to learn a policy that explores, so it will inevitably spend more samples getting to critical states
There are other state of the art baselines that also could have been included like simple behavior cloning or [1].

The MuJoCo experiments could have been augmented to be sparse, like ant maze. Showing that the algorithm matches the performance of HySAC does not provide useful information.

One big concern is the lack of citation of [2]. They study the benefits of uniform simulator resets in terms of sample efficiency and robustness. Findings in 5.1 and 5.2 are somewhat overlapping with the aforementioned work.

[1] Siddhant Haldar, Vaibhav Mathur, Denis Yarats, Lerrel Pinto "Watch and Match: Supercharging Imitation with Regularized Optimal Transport"
[2] Aravind Rajeswaran, Kendall Lowrey, Emanuel Todorov, Sham Kakade "Towards Generalization and Simplicity in Continuous Control"

**Questions:**

How does this algorithm perform in a sparse continuous control task like ant maze?

How does this algorithm perform in higher-dimensional systems like humanoid?

In section 5.3, how do the baseline algorithms perform given the same number of samples (0.5k) as given to AuxSS?

**Details Of Ethics Concerns:**

May not be intentional, but many ideas are similar to an un-cited work [1]

[1] Aravind Rajeswaran, Kendall Lowrey, Emanuel Todorov, Sham Kakade "Towards Generalization and Simplicity in Continuous Control"

---

> ### Author Response · Authors · 2024-11-30
> **Thank you for your time and feedback**
>
> ## **May not be intentional, but many ideas are similar to an un-cited work  [1]. They study the benefits of uniform simulator resets in terms of sample efficiency and robustness. Findings in 5.1 and 5.2 are somewhat overlapping with the aforementioned work.**
>
>
> We thank the reviewer for pointing out this work, as we hadn’t seen it before. We would like to highlight that our work significantly differs from the mentioned work and does not resemble [1], apart from using a robust benchmark to evaluate our findings. Our central contribution is showing that smartly choosing the sampling distribution over a set of potential start states results in more sample-efficient and more performant policies. In contrast, [1] notes that training from a broader start state distribution yields robust policies. It does not delve into what this broader set of start states should be or how they should be sampled. Instead the central premise of [1] is to show the competence of simple linear policy parameterizations for continuous control tasks, which is wholly different from our focus of efficiently bootstrapping online reinforcement learning with as little offline data as possible. We would like to thank the reviewer again for making us aware of this paper, as it helps further reinforce our motivations for robustly evaluating methods. We have added this citation in Section 3, where we motivate the use of the robust benchmarks to evaluate our method.
>
> As an additional technical difference, we point out that the term “robustness” has different meanings in [1] and in our work: In [1], robustness refers to good performance under perturbations of the transition function $p(s_{t+1} | s_t, a_t)$, while in our case, the term robustness refers to good performance under the same transition function but with an altered start state distribution $p_0$. This is akin to the contrast between robustness under drift and under a covariate shift, which are different technical challenges.
>
> [1] Aravind Rajeswaran, Kendall Lowrey, Emanuel Todorov, Sham Kakade "Towards Generalization and Simplicity in Continuous Control"
>
> ## **Unfortunately the baselines in the main experiments are not designed for and do not have access to resetting to arbitrary states and are not directly comparable. JSRL seems to learn a policy that explores, so it will inevitably spend more samples getting to critical states. Showing that the algorithm matches the performance of HySAC does not provide useful information.**
>
> The use of both HySAC and JSRL as strong baselines is motivated by the fact that they are hybrid RL methods that have access to more kinds of privileged information than the proposed method. They access all three of expert state, action and reward (JSRL has been further augmented by us to leverage resetting). In contrast, the proposed approach simply relies on expert state information and resetting. We believe that this comparison provides useful information as showing the proposed method to outperform these baselines demonstrates the effectiveness of resetting at overcoming the lack of expert action and reward information.
>
>
> Note here that, JSRL is a method that is naturally compatible with reset setting, as the handover point between the guide policy and exploration policy can be considered to be the reset point. By only counting the steps post the handover point, we imbue the ability of state resets into JSRL.
>
> For a more direct comparison, the discussion in Section 5.5 and 5.6 explores a direct comparison with other start state distributions. Two of these have been inspired from Go-Explore (Uniform-SS) and JSR (Goal-SS). While these methods do not focus on start state distributions explicitly, they implicitly induce particular distributions over the start state which as we show can impact performance and sample-efficiency.
>
> Also, we would like to highlight that the proposed method learns an exploration policy in the same way as described in JSRL. An exploration policy in JSRL is a policy that is learnt to maximize a task oriented reward from online interaction. It uses exploration as a mechanism to find a higher reward solution for the task. This is exactly the framework under which the proposed method operates. This contrasts the guide policy in JSRL, which is learnt offline and not updated during the online learning stage.

---

> ### Author Response · Authors · 2024-11-30
> **Rebuttal Continued**
>
> ## **There are other state of the art baselines that also could have been included …..**
>
> We have broadened the set of evaluated baselines to include imitation learning methods including behaviour cloning, AIRL and GAIL [3] in Figure 12 within the Appendix. We have evaluated these methods on the simpler Lava Bridge Maze environment having a low dimensional state space and found that none of the methods are able to solve such hard exploration safety critical tasks.. We hope this helps shed further insight.
>
> [2] Justin Fu, Katie Luo, and Sergey Levine. Learning robust rewards with adverserial inverse reinforcement learning. ICLR, 2018.
>
> [3] Jonathan Ho and Stefano Ermon. Generative adversarial imitation learning. NeurIPS, 2016.
>
> ## **How does this algorithm perform in a sparse continuous control task like ant maze? How does this algorithm perform in higher-dimensional systems like humanoid?**
>
> We thank the reviewer for highlighting this.We would like to point the reviewer to the LavaBridge environment which is a sparse reward, continuous action task and results for it can be found in Figure 3 and 12.  In addition, we have augmented our experiments with more sparse reward environments that have a high dimensional state space. Instead of AntMaze and Humanoid which are still relatively low dimensional, we have created two 3D Navigation environments in the Miniworld simulator to directly learn a policy that goes from image to action in a sparse reward setup. We have included these results in Section 5.2, where consistent with previous findings we see that AuxSS outperforms competing methods on sample-efficiency and performance. Of particular note is the performance on the Hard exploration instantiation (see Fig 11 in the appendix) where AuxSS is the only method that can solve the task from the original state distribution.
>
>
> ## **The MuJoCo experiments could have been augmented to be sparse, like ant maze.**
>
> The choice to keep MuJoCo experiments with dense reward tasks is deliberate. These settings do not play to the strengths of the proposed approach. The dense reward provides strong consistent feedback for optimizing the policy by reducing the exploration burden. This reduced exploration burden manifests in the form of rapid saturation of early termination (see Appendix 7.9) which limits the ability of AuxSS to form a targeted distribution. By considering this setting, we show that even in an easy exploration and dense reward setup, the design choice of resetting from expert demonstration states, even with a suboptimal (near uniform) distribution is beneficial due to the presence of rare yet important (task critical) states that are present in the expert demonstration states.
>
> ## **Showing that the algorithm matches the performance of HySAC does not provide useful information.**
>
> The use of HySAC as a strong baseline is motivated by the fact that it is a hybrid RL method that has access to all three of expert state, action and reward. In contrast, the proposed approach simply relies on expert state information and resetting. We believe that HySAC provides a useful comparison as showing the proposed method to outperform this strong baseline demonstrates the effectiveness of resetting at overcoming the lack of expert action and reward information. For more direct comparisons within the resetting regime, we compare with 3 different distributions in Section 5.5 and 5.6. Two of these, GoalDist-SS and Uniform-SS are inspired from JSRL and Go-Explore, where these distributions manifest implicitly in the manner they have formulated the problem.
>
> ## **In section 5.3, how do the baseline algorithms perform given the same number of samples (0.5k) as given to AuxSS?**
>
> Apologies for the confusion. To clarify, besides Figure 6,, the other figures containing results on Lava Bridge, all contain baselines that are computed with 0.5K samples. In particular, Figure 3 contains the comparison of baseline algorithms having access to 0.5K samples with AuxSS when it has access to 0.5K samples.
>
> To better convey this we will further improve the wording for the camera ready if accepted.

---

> ### Author Response · Authors · 2024-12-03
> **Update to IQL+JSRL in Figure 4 and Figure 11**
>
> We would like to share that we completed running IQL+JSRL (5 seeds) on both 3D navigation environments for 10 million steps and continue to see it underperform AuxSS. We will update these figures in the final version of the paper. Below we share the high level trends -
> - On the easy instantiation of the task, IQL+JSRL achieves a max $p_0$ success rate of $80$% vs $100$% achieved by AuxSS. Additionally, its out-of-distribution success rate remains less than $50$% of AuxSS
> - On the hard instantiation  of the task, IQL+JSRL also fails to solve the task from the initial start state distribution while AuxSS completes the task

---

> > ### Author Response · Authors · 2024-12-04
> > **Updated ablations with 3 seeds per experiment**
> >
> > We would like to share that we improved the trustworthiness of the ablations over the demonstration size (Fig 13) and episode horizon (Fig 14) by running each experiment with 3 seeds. The updated figures will be included the final version of the paper. We find that in both cases AuxSS is insensitive to changes in these parameters and the performance of AuxSS remains nearly the same.
> >
> > - For the ablation over demonstration size this makes sense since AuxSS is dependent on the completeness / quality of the episode rather than the quantity of demonstration transitions.
> >
> > - For the ablation over episode horizon, we believe that despite the increase in episode horizon, the episode horizon isn't big enough to flatten out the sampling distribution and it maintains enough "peakiness" in the early stages of training to incentives sampling start states near episode terminations.

---

### Meta-Review · Area_Chair_uNhM · 2024-12-21

**Metareview:**

This paper studies the use of auxiliary start state distributions informed by safety cues to accelerate online reinforcement learning, showing improvements in sample efficiency and robustness using a novel sampling approach. It reports results from experiments on sparse reward and continuous control tasks, including 2D and high-dimensional settings. Despite the interesting concept, the reviewers identified significant concerns, including a lack of direct comparability with state-of-the-art baselines, insufficient exploration of diverse environments, unclear methodological choices, and limited reproducibility due to a low number of seeds in experiments. The reviewers were not convinced during the rebuttal phase, as key questions about experimental design, baseline inclusion, and clarity of the proposed method remained unresolved.

**Additional Comments On Reviewer Discussion:**

During the rebuttal period, reviewers raised several points of concern. Reviewer SGVK highlighted issues with baseline comparability, citing the lack of consideration for methods designed for arbitrary resets and suggesting inclusion of other relevant works. They also questioned the relevance of dense reward MuJoCo experiments and sparse task performance. Reviewer e98q flagged unclear notation, insufficient experimental seeds, and unclear visualizations, particularly regarding the sampling distributions and normalization. Reviewer 1Utq requested evaluations on higher-dimensional tasks with sparse rewards and raised questions about the algorithm's applicability to dynamic or random environments and task horizon considerations. The authors provided clarifications, updated the draft to address notational and visualization issues, added results on a new high-dimensional sparse reward task, and included additional baselines. Despite these updates, critical concerns such as insufficient reproducibility, lack of fair baseline comparisons, and limited scope of environments remained unresolved. The decision weighed the novelty of the idea against the unresolved concerns.

---

### Decision · Program_Chairs · 2025-01-22

Reject